# Intelligent Diagnosis of Thyroid Ultrasound Imaging Using an Ensemble of Deep Learning Methods

**DOI:** 10.3390/medicina57040395

**Published:** 2021-04-19

**Authors:** Corina Maria Vasile, Anca Loredana Udriștoiu, Alice Elena Ghenea, Mihaela Popescu, Cristian Gheonea, Carmen Elena Niculescu, Anca Marilena Ungureanu, Ștefan Udriștoiu, Andrei Ioan Drocaş, Lucian Gheorghe Gruionu, Gabriel Gruionu, Andreea Valentina Iacob, Dragoş Ovidiu Alexandru

**Affiliations:** 1PhD School Department, University of Medicine and Pharmacy of Craiova, 200349 Craiova, Romania; corina.vasile93@gmail.com; 2Department of Pediatric Cardiology, County Clinical Emergency Hospital of Craiova, 200642 Craiova, Romania; 3Faculty of Automation, Computers and Electronics, University of Craiova, 200776 Craiova, Romania; anca.udristoiu@edu.ucv.ro (A.L.U.); stefan.udristoiu@edu.ucv.ro (Ș.U.); andreea.iacob@edu.ucv.ro (A.V.I.); 4Department of Bacteriology-Virology-Parasitology, University of Medicine and Pharmacy of Craiova, 200349 Craiova, Romania; ancaungureanu65@yahoo.com; 5Department of Endocrinology, University of Medicine and Pharmacy of Craiova, 200349 Craiova, Romania; 6Department of Pediatrics, University of Medicine and Pharmacy of Craiova, 200349 Craiova, Romania; cgheonea@gmail.com (C.G.); drcarmen88@yahoo.com (C.E.N.); 7Department of Urology, University of Medicine and Pharmacy of Craiova, 200349 Craiova, Romania; andrei_drocas@yahoo.com; 8Faculty of Mechanics, University of Craiova, 200512 Craiova, Romania; lgruionu@gmail.com; 9Department of Medicine, Indiana University School of Medicine, Indianapolis, IN 46202, USA; gruionu@gmail.com; 10Department of Medical Informatics and Biostatistics, University of Medicine and Pharmacy of Craiova, 200349 Craiova, Romania; dragosado@yahoo.com

**Keywords:** thyroid disorders, ultrasound image, deep learning, neural networks

## Abstract

*Background and Objectives*: At present, thyroid disorders have a great incidence in the worldwide population, so the development of alternative methods for improving the diagnosis process is necessary. *Materials and Methods*: For this purpose, we developed an ensemble method that fused two deep learning models, one based on convolutional neural network and the other based on transfer learning. For the first model, called 5-CNN, we developed an efficient end-to-end trained model with five convolutional layers, while for the second model, the pre-trained VGG-19 architecture was repurposed, optimized and trained. We trained and validated our models using a dataset of ultrasound images consisting of four types of thyroidal images: autoimmune, nodular, micro-nodular, and normal. *Results*: Excellent results were obtained by the ensemble CNN-VGG method, which outperformed the 5-CNN and VGG-19 models: 97.35% for the overall test accuracy with an overall specificity of 98.43%, sensitivity of 95.75%, positive and negative predictive value of 95.41%, and 98.05%. The micro average areas under each receiver operating characteristic curves was 0.96. The results were also validated by two physicians: an endocrinologist and a pediatrician. *Conclusions***:** We proposed a new deep learning study for classifying ultrasound thyroidal images to assist physicians in the diagnosis process.

## 1. Introduction

Autoimmunity is related to the pathogenesis of many thyroid diseases, including hyperthyroidism Graves’ disease, hypothyroidism with autoimmune or Hashimoto’s thyroiditis, asymptomatic and postpartum thyroiditis, and some forms of neonatal thyroid dysfunction [1]. The prevalence of AITD increases exponentially and is more frequent in females. It’s the main cause of addressability to the doctor in the field of pediatric endocrinology and endocrinology [2]. Autoimmune hypothyroidism (AH) is usually divided into goiter (Hashimoto’s thyroiditis (HT)) and non-thyroid primary edema. HT is characterized by extensive lymphocytic infiltration of the thyroid gland, usually accompanied by the formation of germinal centers, while in myxoedema, progressive fibrosis and gland atrophy have almost no inflammatory infiltration.

Autoimmune hyperthyroidism or Graves disease (GD) affects approximately 2% of women and 0.2% of men. The disease is characterized by the presence of thyroid-stimulating antibodies (TSAb) that target the thyroid-stimulating hormone receptor (TSHR) and act as agonists, leading to chronic hyperstimulation and thyrotoxicosis [3].

Increasingly, ultrasound has been used for thyroid structure assessment. It is a valuable addition to the clinical test to determine the size and anatomy of the thyroid and to identify nodules. Diffuse echogenicity reduction (hypoechoic) or the presence of micronodules are frequently observed results in AITD and are considered to be accurate AITD predictors [4]. Higher thyrotropin (TSH) values were observed in pediatric patients with abnormal US findings [5], indicating that the ultrasound may also be beneficial as a non-invasive method to evaluate children for thyroid dysfunction. The ultrasound (US) is far more sensitive to the identification of nodules than physical examination [6]. The diagnosis and treatment of thyroid disease have become significant because of its vital role within the human body.

The nodule was characterized as a distinct hypo-, hyper-, or isoechoic focal area within the thyroid gland described in the US, with different vascularity from the surrounding parenchyma. Lesions with irregular margins often are identified as nodules if they were sonographically differentiated from the adjacent parenchyma. Pseudonodule was characterized as an imprecise hypo-or hyper-echoic focal region [7]. A cyst was described as an anechoic focal area with no solid element.

The conventional diagnostic method, based on the professional expertise of the physicians, has a major deficiency in that the performance of the diagnosis relies primarily on the personal skill and understanding of the physician. Therefore, the accuracy of the diagnosis is limited and depends on the experience of the doctor [8].

The rapid development of ultrasound techniques determined their use as an alternative method for the diagnosis and follow-up of thyroid nodules due to their real-time and noninvasive features. The diagnosis performance can be improved using computer-aided tools for the automatic classification of thyroid nodules.

Deep learning as a subdomain of machine learning tool rapidly evolved in medical imaging analysis and computer vision [9,10,11] and is often considered an alternative tool for analyzing and classifying US images. Previous researchers have demonstrated various approaches to detect nodules in ultrasound images [12].

In [13], it was proposed a method for kidney diagnosing based on convolutional neural networks used to extract instance-level features from 2D US kidney images and graph convolutional networks used to improve them by exploring their correlations.

A comparison between diagnosis performances of deep learning methods and radiologists for differentiating thyroid nodules in ultrasonography was proposed in [14]. Convolutional neural networks recorded comparable performances to radiologists.

In [15], the fusion of two pre-trained CNN was proposed on a large dataset of US images of thyroid nodules. The study results demonstrated that the proposed deep learning methods can diagnose thyroid nodules.

Liu et al. [16,17] proposed an alternative method by integrating high-level features extracted from CNNs and hand-designed low-level features. A pre-trained convolutional neural network with transfer learning was used to generate semantic deep features that were combined with conventional features such as histogram of oriented gradient and scale invariant feature transform.

In [18], was proposed a study based on an ultrasound image-based diagnosis of malignant thyroid nodule and artificial intelligence. The outputs of multiple CNN models were mixed using bagging method rules to improve the classification performance.

In [19] was proposed a DCNN model for identifying thyroid cancer patients, which showed a better performance than skilled radiologists.

The deep learning methods with transfer learning have become valuable tools in medical applications because of their success in minimizing the training time [20]. Additionally, they require fewer data to train, while enhancing the classification performance. However, the question raised was how a pre-trained CNN network could be used to classify medical images, which are very different from the original trained images. Therefore, in this paper, we proposed a fusion between a pre-trained model fine-tuned with our train dataset and an end-to-end trained CNN model. We analyzed their classification performance as separate models and as an ensemble. The advantage of an ensemble method was the reduction of variance by training two models instead of a single model and by combining the predictions of the models.

The contributions of our research consist of:
Designing, developing, training, optimizing, and evaluating a novel fusion method (called CNN-VGG) based on two DL models, in order to increase the classification accuracy.Fine-tuning of pre-trained models for feature extraction and image classification.Designing, developing, evaluating, and optimizing an efficient 5-CNN model with five convolutional layers, in order to analyze its classification abilities.Collecting and curing a novel dataset of 2797 images, which included thyroidal US images, classified into four diagnoses: autoimmune, micro-nodular, nodular, and normal.A detailed experimental and statistical analysis of the proposed models was provided to validate the performance of the proposed methods: accuracy, sensitivity, specificity, positive and negative predictive values, ROC-AUC, and Precision/Recall were taken into consideration.


## 2. Materials and Methods

### 2.1. Patients Data

This multicenter and retrospective research study was conducted at four tertiary referral institutions of Craiova (Craiova Endocrinology Private Clinic, Pediatric Clinic of County Hospital of Craiova, Endocrinology Clinic of the Municipal Hospital of Craiova and Faculty of Automation, Computers and Electronics of Craiova) with study cohorts made up of patients who visited each institution between 2018 and 2020. We had a total of 230 patients of which 30 were children and 200 adults, who were screened for thyroid disorders (ultrasound screening and blood tests).

Approval was obtained from the institutional review boards of all institutions, and requirement for informed consent was obtained as the study design was based on a prospective research of medical tests and ultrasound images (University of Medicine and Pharmacy of Craiova, Craiova Endocrinology Private Clinic, Pediatric Clinic of County Hospital of Craiova, Endocrinology Filantropia Clinic). Informed written consent was obtained from all subjects. The study protocol was approved by the Research and Ethics Committee of the University of Medicine and Pharmacy of Craiova (No.6/20.01.2021) and carried out under the Code of Ethics of the World Medical Association (Declaration of Helsinki) for experiments involving humans. All experiments were performed in accordance with relevant guidelines and regulations. All images and pathologic data were anonymized before being transferred between different hospitals.

Thyroid ultrasounds were performed by endocrinologists with more than 20 years of significant experience, using a low-medium-resolution (7.5 MHz) ultrasound instrument (Siemens SonoAce) with a linear transducer. The patient remains comfortable during the test, which typically takes only a few minutes unless there is a need to evaluate the lateral neck, does not require discontinuation of any medication, or preparation of the patient. The procedure is usually done with the patient reclining with the neck hyperextended but it can be done in the seated position. For diagnostic purposes, multiple laboratory tests were performed to assess thyroid function: TSH (thyroid-stimulating hormone), free T4 (thyroxine), presence of thyroid antibodies: thyroid peroxidase antibodies, thyroglobulin antibodies, blood glucose levels, calcium, liver function, vitamin D dosage. Thyrotropin Receptor Antibodies (TRAb) were further dosed in patients diagnosed with hyperthyroidism. Ultrasound features were noted such as thyroid volume, ultrasound pattern, heterogeneity, nodule presence and size, the existence of pseudo-nodules and cysts.

For each patient, 4 ultrasound images were interpreted, in cross and longitudinal sections for each lobe. We collected 920 patients’ images split into four diagnosis: autoimmune (260), micro-nodular (215), nodular (242), and normal (203) (Figure 1).

### 2.2. Image Dataset

The initial dataset of 920 images collected from 230 patients was augmented by the following transformations: vertical and horizontal translations, random shearing and random zooming transformations. Therefore, the final augmented dataset contains 2797 grayscale thyroid images in gray-scale with an initial resolution of 500 × 500 pixels, after cropping the images borders and patient information. We resized the images to a resolution of 224 × 224 pixels. Of those, 185 patients with 2297 images were used for training and 45 patients with 500 images for testing. The distribution of patients and images among the diagnosis is described in Table 1.

### 2.3. Convolutional Neural Networks

Convolutional neural networks (CNNs) are a type of artificial neural network and one of the most powerful tools for computer vision applications.

The base process of CNNs is a convolution operation where the input image is convolved using filters that detect important features of an image. The network is capable of automatically learn the filter’s value that detects the patterns to match the wanted output, such as the diagnosis of the input image. The learning process is realized through an activation function that makes possible the back-propagation technique, by calculating the error between the predicted and real values of the data. Then, this error is propagated throughout the network, changing the weights of the filters. Another input image is fed to the network and the learning process is repeated iteratively improving the algorithm. The activation function determines the output of each convolution process and reduces the complexity of the neural network [21].

The operation of a convolutional layer is given by (1):*FM*(*i*,*j*) = (*I* * *F*)(*i*,*j*) = *∑∑I*(*i* + *m*, *j* + *n*)*F*(*m*, *n*)(1)
where *I* represents the input matrix, *F* refers to a 2D filter of size (*m*, *n*), *I * F* denoted the convolutional operation and *FM* represents the output of a 2D feature map.

The convolutional layers are alternated with pooling, batch normalization and fully connected layers (FC). A pooling layer follows a convolution layer to down-sample the feature map of the convolution layer. The batch normalization layers are used to normalize the input layers. In a dropout layer, the neurons are randomly disabled to reduce overfitting and force the model to learn multiple independent representations of the same data [22].

### 2.4. Transfer Learning with Fine Tuning

Since most CNNs construct the convolution layers deeper and deeper to achieve better performance, the pre-trained CNN models with transfer learning were a significant milestone in the development of deep learning classifiers. Very deep CNN networks have a risk of overfitting, while the pre-trained networks are complex architectures that use improvements to obtain better efficiency.

Transfer learning is a method of training, which uses an existing pre-trained classifier as a starting point for a new classification task [20]. Deep neural models (AlexNet, VGG, ResNet, and Inception) trained on large-scale datasets such as ImageNet have recorded very good results with transfer learning. These networks can learn a set of discriminating features to recognize 1000 object classes.

Fine-tuning is a type of transfer learning. The fine-tuning is applied to DL models that have already been trained on a given dataset. This method consists of removing the final set of fully connected layers of the pre-trained network, and replace them with a new set of fully connected layers with random initializations [20].

### 2.5. Deep Models Implementation

#### 2.5.1. The 5-CNN Model

The first method we proposed, called 5-CNN, was an efficient lightweight architecture based on CNN network and trained in an end-to-end manner on the train dataset of 2297 US thyroidal images. The proposed network had 11 layers: 5 convolutional layers, 4 pooling layers, one fully connected layer, and one output layer with the softmax function [21]. The proposed 5-CNNs model was described in Figure 2.

The algorithm steps for defining and training the 5-CNN model were described in Algorithm 1:
**Algorithm 1. The 5-CNN model description****Input**: thyroidal images of dimension (500 px, 500 px) from the train dataset.**Output**: CNN model weights1. **for** each image in the dataset 2. Resize image to (224 px, 224 px) 3. Normalize the image pixels values between [0, 1]. 4. **end**
5. Add a first convolutional layer with a RELU activation function. 6. Add a second convolutional layer with a RELU activation function. 7. Apply a max pooling layer for down-sampling feature map from the previous layer.8. Repeat the steps 4 and 5 for three times. 9. Add a Flatten layer on the output obtained from the last max-pooling layer. 10. Add a fully connected layer with 256 hidden units. 11. Apply a dropout for inactivate neurons in the previous layer. 12. Add a fully connected layer with 4 hidden units and a softmax activation function. 13. Optimize the model with RMSProp optimizer with a learning rate of 0.0001. 14. Train the model for 100 epochs. 15. Save the final model.

#### 2.5.2. The VGG-19 Model

The pre-trained models used in this study were VGG-19 [23], ResNet50 [24], Inception-v3 [25], and EfficientNetB0 [26], which were all pre-trained on ImageNet dataset [23] that includes non-medical images. Each model was fine-tuned with 2297 images from the train dataset.

Based on the results of each pre-trained model on the test dataset, we chose the VGG-19 model, which recorded the best classification performance. The other models had weaker performance in classifying our US thyroidal images due to the small size of the dataset and lower quality of the images.

VGG-19 is a variant of the VGG model composed of 19 layers from which, 16 layers are convolution layers, 3 layers are fully connected, 5 layers are maximized pool and 1 layer is a softmax layer [23]. As we repurposed the pre-trained VGG-19 model for medical diagnosis classification, we removed the original classifier, added a new classifier that fits our purposes, and finally, we had to fine-tune our model. Our strategy for fine-tuning was to freeze the convolutional base and then used its outputs to feed the diagnosis classifier [27]. At the top of the network, we added one fully connected layer of 256 hidden units and ReLU activation. The last layer was the softmax dense layer used for classification. The model was optimized by using the Adam optimizer with a learning rate of 0.0001 [28]. The proposed VGG-19 model with transfer learning and fine-tuning was described in Figure 3.

The algorithm steps for defining and training the VGG-19 model were described in Algorithm 2:
**Algorithm 2. The VGG-19 model description****Input**: thyroidal images of dimension (500 px, 500 px) from the train dataset.**Output**: VGG model weights 1. **for** each image in the dataset 2. Resize image to (224 px, 224 px) 3. Normalize the image pixels values between [0, 1]. 4. **end**
5. Load the VGG-19 model pre-trained on ImageNet dataset. 6. Remove the last layer of the model. 7. Make non-trainable all the layers of the model. 8. Add a Flatten layer on the model output to obtain a 1-D array of features. 9. Add a fully connected layer with 256 hidden units. 10. Apply a dropout for inactivate neurons in the previous layer. 11. Add a fully connected layer with 4 hidden units and a softmax activation function.  12. Optimize the model with Adam optimizer. 13. Train the model for 100 epochs. 14. Save the final model.

#### 2.5.3. The CNN-VGG Ensemble Method

In order to increase the classification performance, we designed and implemented an ensemble method called CNN-VGG, which fused the VGG-19 and 5-CNN models with their output predictions using averaging to as in Figure 4.

The probabilities of the two trained models (5-CNN and VGG-19) were averaged to generate new probabilities (P_n_) for the final diagnosis decision as in (2):P_n_ = Average (P_1n_ + P_2n_)(2)
where n = 1.4.

The Jensen’s inequality proves that the average ensemble will have the error less than or equal to the average error of the individual models [29]. The combinations of predictions from multiple convolutional neural networks added a bias, but also reduced the variance of a single model. The resulting predictions were less sensitive to the particularities of the training dataset.

The algorithm steps for the proposed ensemble were described in Algorithm 3:
**Algorithm 3. The VGG-CNN ensemble description.****Input**: thyroidal images of dimension (500 px, 500 px) from the test dataset.**Output**: prediction probabilities for each diagnosis class (autoimmune, micro-nodular, nodular, normal). 1. **for** each image in the dataset 2. Resize image to (224 px, 224 px) 3. Normalize the image pixels values between [0, 1]. 4. **end**
5. Load the trained 5-CNN model. 6. Load the trained VGG-19 model. 7. Predict the images with CNN resulting a list of probabilities (P11, P12, P13, P14) 8. Predict the images with VGG resulting a list of probabilities (P21, P22, P23, P24) 9. Average the two lists of predictions of the two models. 10. **for** each class in the set of diagnosis 11.Output prediction probabilities for the diagnosis class. 12. **end**

### 2.6. Evaluation Metrics

The performance metrics used to evaluate our methods were: specificity (Sp) (3), sensitivity or recall (Se) (4), negative positive values (NPV) (5), positive predictive values (PPV) (6), test accuracy (Accuracy) (7).
Sp = TrueNegatives/(TrueNegatives + FalsePositives)(3)
Se = TruePositives/(TruePositives + FalseNegatives)(4)
NPV = TrueNegatives/(TrueNegatives + FalseNegatives)(5)
PPV = TruePositives/(TruePositives + FalsePositives)(6)
Accuracy = CorrectlyClassifiedCases/TotalCases(7)

Other tools used to evaluate the quality of the classifier were the receiver operating characteristic (ROC) and Precision/Recall (P/R) curves [30]. We also computed the micro averaging area under the curve (ROC-AUC) and micro averaged precision to evaluate the overall performance across all diagnosis classes. In micro-averaging, we computed the average of the separate true positives, true negatives, false positives, and false negatives of each diagnosis class. Having a multi-class classification problem, the micro-averaging computation for metrics follows the one vs. rest approach. We defined our metrics for each class as actual class (for example, normal class) vs. other classes (autoimmune, micro-nodular, and nodular). For example, for the normal diagnosis class, TP is the number of images correctly predicted as Normal, FP denotes the other cases that are misclassified as Normal by the model, TN denotes the other cases that are correctly classified, and FN denotes the Normal cases that are misclassified as other cases.

## 3. Results

The dataset contains images from 230 patients diagnosed with thyroid disorders of which 30 were children aged between 17 days and 17 years old, and 200 adults aged between 18 years and 75 years old, with a higher predominance of female gender. From the total number of patients, 78 were from rural communities and 152 from metropolitan centers. Hypothyroidism was confirmed in 75 patients, subclinical hypothyroidism in 15 patients, congenital hypothyroidism in 2, hyperthyroidism in 38, subclinical hyperthyroidism in 10, and euthyroidism in 90 patients.

### 3.1. Experimental Setup

The dataset is a 4-classes classification problem having 2797 US images. The dataset was randomly divided into train dataset and test dataset (80% of images for training and 20% of images for testing), using the pattern train-validation-test. The dataset was divided patient-wise. In this study, the validation and test sets are the same, but the images from the test dataset are not used for training.

We designed three experiments:

1. Training and evaluating the 5-CNN model.

2. Training and evaluating the VGG-19 model.

3. Evaluating the CNN-VGG ensemble method.

### 3.2. Deep Models Evaluation and Statistical Analysis

Experiment 1: 5-CNNs model

The 5-CNNs model was trained for 100 epochs on the training dataset (2297 images) with a computational time for an epoch of 80 s. The model was assessed on the test dataset (500 images). From the experimental results summarized in Table 2, it could be observed that the 5-CNNs model had a good ability to distinguish the autoimmune, and normal images from the test dataset. The classification results for micro-nodular and nodular images were under the average results.

Experiment 2: VGG-19 model

We trained the VGG-19 model on the training dataset (2297 images) for 100 epochs. The computational time for training the VGG-19 model was of 70 s per epoch. The results were obtained on the test dataset (500 images) and showed an overall diagnosis accuracy of 96.7% and an AUC of 0.95. The good performance in terms of sensitivity and specificity for all diagnosis showed that the proposed VGG-19 method had a good generalization in classifying thyroidal images.

The accurate results obtained using the VGG-19 model with transfer learning were summarized in Table 3.

Experiment 3: CNN-VGG ensemble method

The experimental results were obtained on the test dataset (500 images) and showed that the CNN-VGG ensemble method outperformed each model with an increase of 1–5% in all metrics. The results are summarized in Table 4.

When using the CNN-VGG ensemble method, the autoimmune diagnosis class was classified with high sensitivity and specificity (97.6%, and 99.26%, respectively). The values of sensitivity (97.6% for autoimmune cases, 92.8% for micro-nodular cases, 92.61% for nodular cases, and 100% for normal cases, respectively) mean that the sum of the false negatives is low while the specificity values (99.26% for autoimmune cases, 96.57% for micro-nodular cases, 97.89% for nodular cases and 100% for normal cases, respectively) mean that the sum of the true negatives is high.

The maximum PPV (100%) was achieved for the normal cases and a lower value of PPV (93.87%) was obtained using nodular cases. The autoimmune diagnosis achieved the maximum NPV (99.02%), while the minimum NPV was obtained using nodular image cases. The performance of CNN-VGG method increased in recognising the micro-nodular and nodular cases.

The ROC curves of the ensemble CNN-VGG method computed for each diagnosis class showed the rate of false-positive was near zero while the rate of true positive was between 0.9 and 1 (Figure 5).

The Precision/Recall curves of the ensemble method showed good precision and recall for all diagnosis classes of thyroidal images (Figure 6). However, as all metrics indicated, the classification performance for micro-nodular and nodular images was weaker.

The experimental results indicated that the proposed ensemble CNN-VGG method can be a good and stable classifier for thyroidal images.

## 4. Discussion

The main purpose of this research was to achieve good results in detecting patterns of thyroidal US images. By analyzing the results, our proposed ensemble method based on CNN and fine-tuning had significant results in the classification of thyroid nodules images. We tried to construct a stable and solid classifier that combined both the transfer learning and training from scratch, balancing the training of a medium-size medical dataset, but also different from the original ImageNet dataset. The better performance of a model averaging ensemble is because the deep learning models will usually not make the same errors on the test dataset [31].

Additionally, previous studies on diagnosing thyroidal images showed better performance when using ensembles than using CNNs alone. For example, in the study [32] that introduced very deep convolutional neural networks (AlexNet) for photo classification, Alex Krizhevsky et al. used the model averaging of multiple well-performing CNNs. The performance of one model was inferior to the predictions of an ensemble composed of two, five, and seven models.

In his study, Ma et al. [15] proposed a hybrid method for thyroid nodule diagnosis validated on 15,000 ultrasound images with an accuracy of 83.02% ± 0.72%. In the study performed by Liu et al. [16], a feature extraction method was proposed based on the convolution neural networks (CNNs) in order to discriminate benign and malignant nodules in ultrasound images. The experimental results were realized based on 1037 images and recorded an accuracy of 93.1%. Zhu et al. [33] used the ResNet18-model with transfer learning to classify the malignant and benign thyroid nodules and obtained an accuracy of 93.75%. For the same problem, Chi et al. [34] used the Inception CNN model on a public thyroid nodule dataset. The reported accuracies of diagnosing the thyroid nodules were 99.13% and 96.34% for the used testing datasets. Song et al. [35] used the pre-trained Inception v3 model to classify nodules as benign or malignant. They reported a sensitivity of 94% and NPV of 90.3% on the test dataset. In another recent publication [36], a computer-aided detection system was developed for the segmentation of nodules in thyroid US images and their classification as malign or benign. Their results were 93.88 ± 2.59% for Dice Coefficient and 91.18 ± 7.04 pixels for Overlap Metric.

The above-described studies classified the thyroid nodules as malign or benign. To the best of our knowledge, the proposed study was the first attempt of using the deep learning methods in analyzing and classifying multiple classes (autoimmune, micro-nodular, nodular and normal) of thyroidal images.

Our study was supposed a difficult task due to the number of diagnosis we had to classify and the low contrast between the background and thyroid nodules. In order to improve the resolution quality of thyroidal ultrasound images, the super-resolution deep learning methods will be taken into consideration [37,38]. These methods are difficult to be implemented in real time ultrasound due to many network parameters and convolution operations in deep network layers. Therefore, the trade-off between complexity and performance will be taken into consideration into the future.

We are aware that our study had some limitations. First, collecting some representative images from each patient was dependent on the experience and skills of the endocrinologist. It is obvious that the deep learning methods still need human intervention in the process of image collection. In practice, physicians evaluate thyroid patterns using real-time US information, not a specific image.

Second, the dataset known in advance was another issue, because there was a risk of overfitting, even if the train and test datasets are image-independent. While this is an appropriate accuracy rate, future research will evaluate and refine the algorithm on a larger patient database to resolve the current small sample size and lack of an independent test cohort. Third, we will take into consideration a more complex validation of our results in real-time US practice.

In conclusion, the proposed methods showed comparable diagnostic performances to expert endocrinologists in classifying specific patterns of thyroidal US images from the test set.

## 5. Conclusions

In this study, we focused on finding a way to differentiate the features of thyroidal disorders, in order to diagnose the US images. For this purpose, we developed a CNN-VGG ensemble fused from two models: a pre-trained fined tuned model VGG-19 and an efficient lightweight CNN model. The proposed ensemble method proved to be an excellent and stable classifier with a good performance in terms of overall sensitivity (95.75%), specificity (98.43%), accuracy (97.35%), AUC (0.96), positive predictive value (95.41%) and negative predictive value (98.05%).

Although artificial intelligence is not ready to substitute physicians in the coming years, clinical experts can learn both the fundamentals of artificial intelligence innovation and how AI-based structures will help them provide greater benefits for their patients at work. In clinical practice, our deep learning model could assist endocrinologists by offering a second opinion in the process of diagnosis.

## Figures and Tables

**Figure 1 medicina-57-00395-f001:**
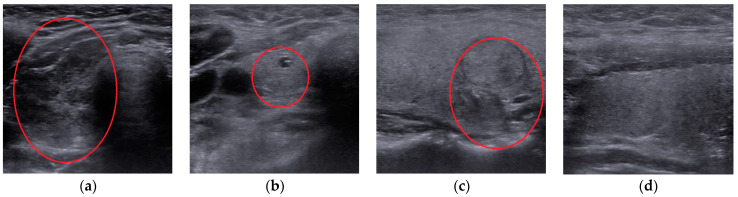
Thyroidal US images with highlighted region of interest: (**a**) autoimmune; (**b**) micro-nodular; (**c**) nodular; (**d**) normal.

**Figure 2 medicina-57-00395-f002:**
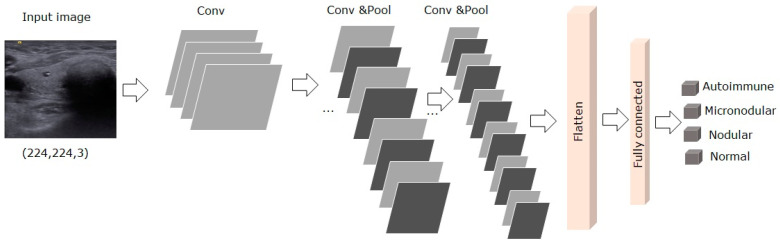
The architecture of the proposed end-to-end trained 5-CNN model.

**Figure 3 medicina-57-00395-f003:**
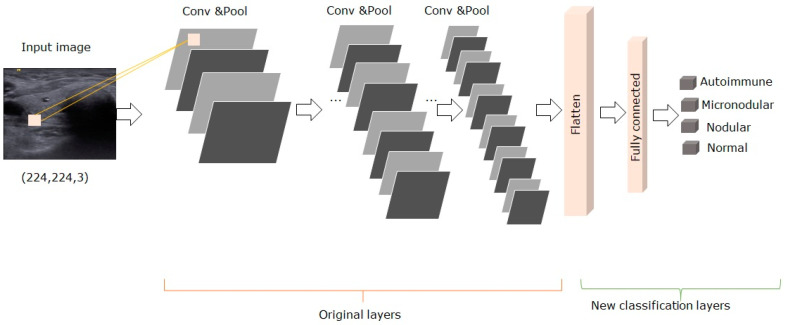
The architecture of the proposed pre-trained VGG-19 model.

**Figure 4 medicina-57-00395-f004:**
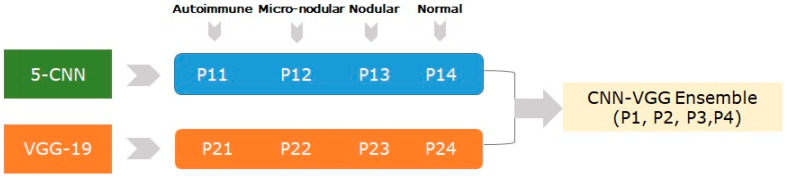
The structure of the CNN-VGG ensemble.

**Figure 5 medicina-57-00395-f005:**
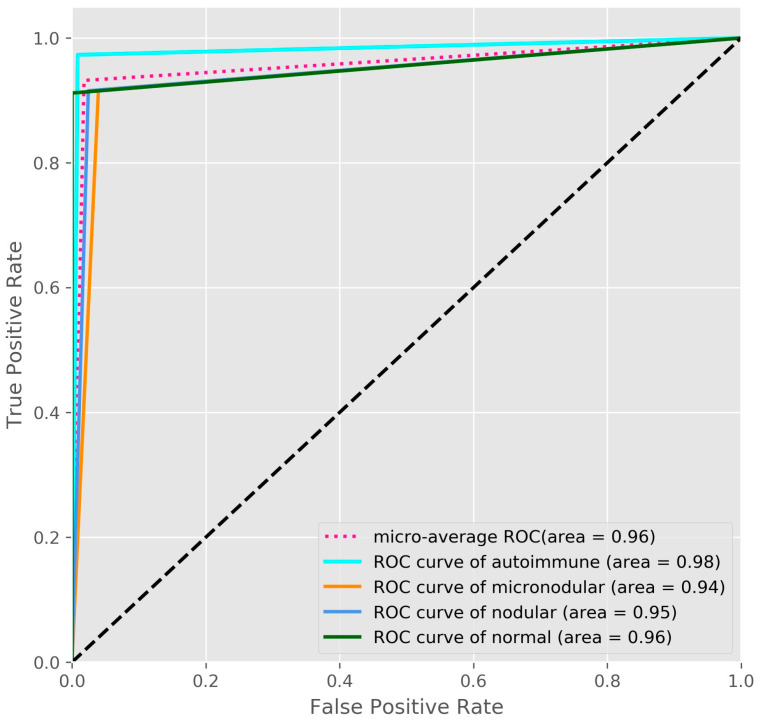
The ROC-AUC curves for the CNN-VGG ensemble method.

**Figure 6 medicina-57-00395-f006:**
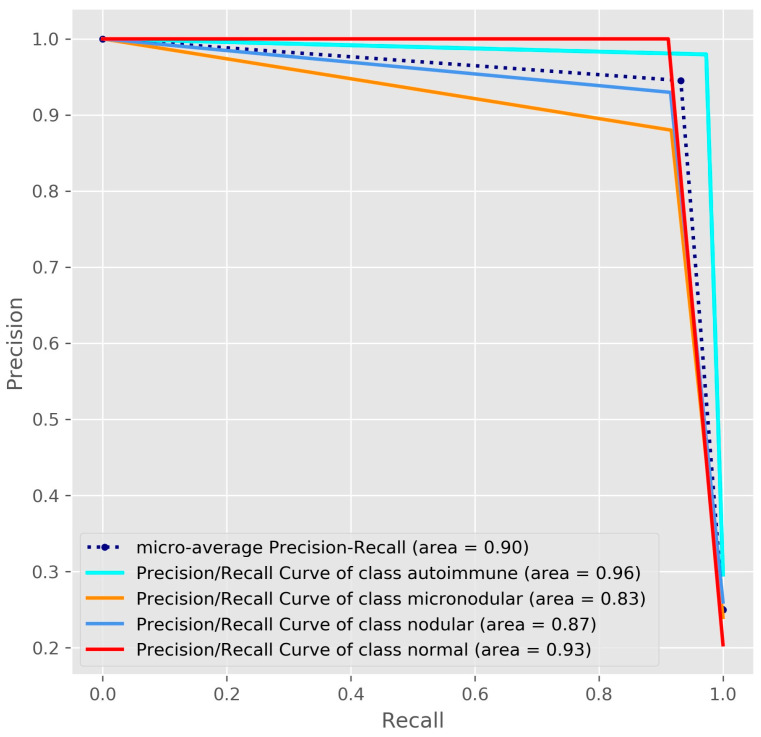
The Precision/Recall curves with averaged precision for the CNN-VGG ensemble method.

**Table 1 medicina-57-00395-t001:** The distribution of images and patients in the training and testing augmented datasets.

Diagnosis	Training	Testing	Totals
Autoimmune	619/67	148/16	767/83
Micro-nodular	552/37	120/9	672/46
Nodular	590/49	130/12	720/61
Normal	536/32	102/8	638/40
Total	2297/185	500/45	2797/230

**Table 2 medicina-57-00395-t002:** Evaluation metrics for classification of thyroid ultrasound images with 5-CNNs model.

Diagnostic Class	Accuracy (%)	ROC-AUC	Sensitivity (%)	Specificity (%)	PPV (%)	NPV (%)
Autoimmune	97.57	0.95	95.2	98.53	96.36	98.05
Micro-nodular	92.89	0.92	91.05	94.74	93.02	95.84
Nodular	92.2	0.92	91.93	92.99	91.70	96.36
Normal	96.88	0.93	95.24	100	100	96.19
Average	94.88	0.93	93.35	96.56	95.27	96.61

**Table 3 medicina-57-00395-t003:** Evaluation metrics for classification of thyroid ultrasound images with VGG-19 model.

Diagnostic Class	Accuracy (%)	ROCAUC	Sensitivity (%)	Specificity (%)	PPV (%)	NPV (%)
Autoimmune	97.2	0.95	90.5	98.71	99.3	96.17
Micro-nodular	95.8	0.97	91.16	94.47	89	99.73
Nodular	95.8	0.94	91.53	97.29	92.24	97.03
Normal	98	0.95	90.19	99.7	98.92	97.54
Average	96.7	0.95	90.84	97.54	94.86	97.62

**Table 4 medicina-57-00395-t004:** Evaluation metrics for classification of thyroid ultrasound images with CNN-VGG method.

Diagnostic Class	Accuracy (%)	ROC-AUC	Sensitivity (%)	Specificity (%)	PPV (%)	NPV (%)
Autoimmune	98.78	0.98	97.6	99.26	98.19	99.02
Micro-nodular	95.66	0.95	92.8	96.57	89.58	97.69
Nodular	96.53	0.95	92.61	97.89	93.87	97.44
Normal	98.44	0.96	100	100	100	98.06
Average	97.35	0.96	95.75	98.43	95.41	98.05

## Data Availability

The data presented in this study are available on request from the corresponding authors.

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
