# Peer review of "Intelligent Diagnosis of Thyroid Ultrasound Imaging Using an Ensemble of Deep Learning Methods"

_medicina, 2021, doi:10.3390/medicina57040395_

Round 1

Reviewer 1 Report

The authors present an interesting study for the classification of thyroid nodules in ultrasound images. They have fused two deep learning models to get accurate classification results.  However, I have the following comments:

The authors stated that ''the dataset was randomly divided into train dataset and test dataset (80% of images for training and 20% of images for testing)''. When working with datasets containing multiple images of the same patient, in my opinion, the splits for cross-validation should be done in such a way so that all images of the same patient fall into the same set (training or test); i.e., the splitting should be based on patients and not on images. Images of the same patient are highly correlated, therefore, using some of them for training and others for testing can lead to optimistic results. Thus, the authors are recommended to split the dataset patient-wise not image-wise. This may change the results presented on the manuscript. If the authors have divided the data patient-wise, they should state that clearly on the manuscript.

In turn, the speckle noise and artifacts that appear in ultrasound images may degrade their performance. To tackle this problem, many related studies have proposed the use of a super-resolution method to enhance the images before feeding them to the feature extraction stage, for instance, ''Breast tumor classification in ultrasound images using texture analysis and super-resolution methods'', ''Super-resolution of retinal images using multi-kernel SVR for IoT healthcare applications'', to name but few. The authors are recommended to consider this issue on their experiments or discuss it in the manuscript.

The authors are recommended to follow the EQUATOR Network guidelines. Specifically, they are highly recommended to complete the Checklist for Artificial Intelligence in Medical Imaging (CLAIM)

https://pubs.rsna.org/doi/pdf/10.1148/ryai.2020200029

For instance, CLAIM-EQUATOR recommends validating or test the models on external data (item 32), however the authors did not consider this important issue on their study. Also, the details of the creation of the ground truth have not been mentioned, for instance, how missing data were handled? and Measurement of inter- and intra-rater variability; methods to mitigate variability and/or resolve discrepancies.

The authors are recommended to rewrite the CNN-VGG Algorithm in the standard format of algorithms. Please complete the definition of NPV in page 8: ''the negative (NPV)''. Please clarify in the text how you calculate the Accuracy, AUC, and other metrics for the Normal class only, meaning how the TP, FP, TN and FN have been computed. The word ''Diagnosis'' appearing in Table 2 and 3 could be replaced by ''class''.

Reviewer 2 Report

This paper presents a deep learning based automated thyroid diagnosis scheme. which is a combination of modified existing models. There are some incremental novelty in the proposed approach, and the topic is of interest to the readers of Medicina. However, the quality of the paper need to be further improved before being considered for publication. My suggestions include:

  1. Besides simply present experimental results showing that the ensembled CNN-VGG method outperforms CNN and VGG separately, more discussion or analysis on the reason why the combination leads to better performance should be provided so that the paper could gain more strength.
  2. The writing of the paper needs improvement. Many sentences, including the title, should be rephrased, e.g., "the question who often raised was how could be adapted ..." in line 94, "185 patients and 2,297 images were used for training and 45 patients, and 500 images for testing" in line 160-161. Besides, abbreviations should be explained at first use, e.g., "US" in line 29.
  3. The Introduction part could be more comprehensive. Recent works on deep learning based diagnosis methods using ultrasound imaging are closely related to the topic of the paper and should be reviewed, e.g., "Multi-instance deep learning with graph convolutional neural networks for diagnosis of kidney diseases using ultrasound imaging" in MICCAI CLIP'19.

Round 2

Reviewer 1 Report

The authors have addressed all my comments. Thanks. The regions of interest can be highlighted in the US images of Figure 1 to improve the understanding of the reader to this figure. Please polish the English of the paper, I still notice typos and grammatical errors in the revised manuscript, for example, ''will be take into consideration'' on page 12. 

Reviewer 2 Report

The authors have adequately addressed the reviewers' concerns, hence I recommend to accept the paper for publication in Medicina.
